# Dopamine Therapy and the Regulation of Oxidative Stress and Mitochondrial DNA Copy Number in Patients with Parkinson’s Disease

**DOI:** 10.3390/antiox9111159

**Published:** 2020-11-20

**Authors:** Shih-Hsuan Chen, Chung-Wen Kuo, Tsu-Kung Lin, Meng-Han Tsai, Chia-Wei Liou

**Affiliations:** 1Department of Neurology, Kaohsiung Chang Gung Memorial Hospital and Chang Gung University College of Medicine, Kaohsiung 83301, Taiwan; b9502054@cgmh.org.tw (S.-H.C.); tklin@adm.cgmh.org.tw (T.-K.L.); menghan@cgmh.org.tw (M.-H.T.); 2Core Laboratory for Phenomics and Diagnostics, Kaohsiung Chang Gung Memorial Hospital and Chang Gung University College of Medicine, Kaohsiung 83301, Taiwan; bulakuo@cgmh.org.tw; 3Center for Mitochondrial Research and Medicine, Kaohsiung Chang Gung Memorial Hospital and Chang Gung University College of Medicine, Kaohsiung 83301, Taiwan

**Keywords:** Parkinson’s disease, oxidative stress, TBARS, thiols, mtDNA copy number, dopamine

## Abstract

Few studies have reported on changes to oxidative stress and mitochondrial DNA copy numbers in patients with Parkinson’s disease (PD), particularly those undergoing long-term dopamine therapy. This study measured mitochondrial copy numbers, thiobarbituric acid reactive substances (TBARS), and thiols in 725 PD patients and 744 controls. The total prescribed dopamine dose was calculated for each PD patient. A decreased mitochondrial copy number and antioxidant thiols level, but an elevated oxidative TBARS level presented in PD patients. Stratification into age subgroups revealed a consistently lower mitochondrial copy number and thiols in all PD subgroups, but increased TBARS levels compared with those of the controls. Further study found an association between lower serum TBARS and dopamine administration. There appears to be an indirect relationship with the mitochondrial copy number, where a decrease in TBARS was found to diminish the effect of pathogenetic and age-related decrease in mitochondrial copy number in PD patients. Follow-up evaluations noted more significant decreases of mitochondrial copy numbers in PD patients over time; meanwhile, dopamine administration was associated with an initial decrease of the TBARS level which attenuated with high-dose and long-term therapy. Our study provides evidence that moderate dopamine dose therapy benefits PD patients through attenuation of oxidative stress and manipulation of the mitochondrial copy number.

## 1. Introduction

Mitochondrial dysfunctions and their consequent generation of oxidative stress have been identified as significant causes of neuronal cell damage, leading to the development of neurodegenerative disease, including sporadic Parkinson’s disease (PD) [1]. Several mitochondria-related disorders affecting different locations within the brain have reportedly been implicated in disease pathogenesis. More specifically, the dopaminergic neurons located within the substantia nigra of the mid-brain are direct targets of the free radical by-products created during the process of energy production, and the by-products of dopamine metabolism [2,3]. Accumulation of injuries from these radicals causes mitochondrial DNA (mtDNA) damage as well as dysfunction of the overall mitochondrial machinery [4,5]. To overcome insufficient cellular energy supply due to this dysfunction, the process of mitochondrial biogenesis is triggered [6]. This process increases the mtDNA copy number and translated respiratory enzymes to sustain normal mitochondrial function in cells. However, failure to increase mitochondrial respiration may render an insufficient energy supply for overcoming cellular stress, eventually leading to the development of disease. Observations of high levels of deleted mtDNA and depletion of wild-type mtDNA in substantia nigra neurons have been observed in brain samples of patients with advanced PD [7,8,9]. Additionally, a decreased mtDNA copy number has been reported in the rapid turn-over peripheral blood cells of PD patients at early stages of the disease [10,11,12]. A serial cascade scenario, involving oxidative stress, mtDNA damage, and mtDNA copy number change, plausibly takes place in cells during the progress of PD. However, as finding direct evidence of this by retrieving tissue samples from living brains is problematic, we thus hypothesized that this serial cascade response may be detected by monitoring changes in human blood samples. We herein report the results of our investigation into the baseline and long-term changes of oxidative stress and the mtDNA copy number, and the probable effects of continuous dopamine therapy on these biomarkers in PD patients.

## 2. Materials and Methods

### 2.1. Subjects

Two groups of Taiwanese subjects of an ethnic Chinese background were recruited into this study. Group 1 consisted of 725 patients (396 male and 329 female) with an average age of 67.1 ± 11.1 (standard deviation) years that had been diagnosed with PD on the basis of a progressive history and more than two of the cardinal signs of PD being present (resting tremor, bradykinesia, rigidity, postural instability). All PD patients were diagnosed clinically by at least one medical doctor specializing in movement disorders in the Neurology Department of Kaohsiung Chang-Gung Memorial Hospital. Patients with secondary parkinsonism or atypical clinical features, such as dementia at an early stage of the disease, pyramidal signs, cerebellar signs, gaze paresis, or significant focal lesions on computerized tomography or magnetic resonance imaging scans were excluded. Group 2 comprised 744 ethnically-, age-, and sex-matched control subjects (403 male and 341 female) with a mean age of 66.3 ± 9.4 years, who had participated in our health examination and had been examined clinically without significant signs of neurological or cognitive impairment related to PD. Written informed consent was obtained from all participants, in accordance with protocols approved by the institutional review board at the Kaohsiung Chang-Gung Memorial Hospital (IRB number: 950615, approved on 25 December 2006; and IRB number: 201601936B0, approved on 23 January 2017). The study was performed in accordance with the Declaration of Helsinki and its text revisions. Venous blood samples were collected after an overnight fast. The DNA was isolated from leucocytes using the PUREGENE^®^ DNA Purification Kit (Gentra, Minnesota, MN, USA). After random periods of between 6 and 64 months, 357 of the PD patients and 392 of the non-PD controls received another follow-up blood collection for study. The main characteristics of these patients and controls are summarized in Table 1. In the PD group, all patients at the time of the initial and follow-up blood collections had been administered dopamine or dopamine agonist. We therefore estimated the total dopamine dose at those times, including the administered dopamine and dopamine agonist for each patient, by using a total equivalent dopamine dose (TEDD) equation [13].

### 2.2. Assessment of Antioxidant and Oxidant Markers

Serum free thiols, an antioxidant parameter, were determined by directly reacting thiols with 5,5’-dithio-bis-(2-nitrobenzoic acid) (DTNB) to form 5-thio-2-nitrobenzoic acid (TNB). The thiols levels were calculated from the absorbance determined using the extinction coefficient of TNB (A412 = 13,600 M^−1^cm^−1^) [14]. The serum thiobarbituric acid reactive substance (TBARS) concentration, which is produced following oxidative stress, was assessed based on the method of Ohkawa et al. [15]. The results are expressed as micromoles of TBARS per liter. A TBARS standards curve was obtained by a hydrolysis of 1,1,3,3-tetraethoxypropane (TEPP).

### 2.3. Measurement of Leukocyte mtDNA Copy Number

The relative mtDNA copy numbers were measured by a real-time PCR and corrected by simultaneous measurement of the nuclear DNA. The forward and reverse primers for a nuclear gene, which are complementary to the β-actin gene, were 5′-TCACCCACACTGTGCCCATCTACGA-3′ and 5′-CAGCGGAACCGCTCATTGCCAATGG-3′. The forward and reverse primers for mtDNA, which are complementary to the sequence of the ND1 gene, were 5′-TGGGTACAATGAGGAGTAGG-3′ and 5′-GGAGTAATCCAGGTCGGT-3′. The PCR was performed in the ABI PRISM 7700 Sequence Detection System (PE Biosystems, Foster City, CA, USA), using the SYBR^®^ GREEN PCR MASTER MIX Kit (Applied Biosystems, Sparta, NJ, USA). The melting curves analysis was provided by the Dissociation Curve Software. The threshold cycle number (Ct) values of the β-actin gene and the mitochondrial ND1 gene were determined for each individual in the same quantitative PCR run. Each measurement was carried out at least three times and normalized in each experiment against a serial dilution of a control DNA sample. Generally, further measurements were requested if samples could not meet the criteria of standard deviation less than 0.1. If no acceptable results were acquired, we gave up the sample and repeated with blood from another sample collection. Patient data exhibiting unusually low or high levels were discarded. Ct values can be used as a measure of the input copy number and Ct value differences used to quantify the mtDNA copy number relative to the ß-actin gene with the following equation: Relative copy number (Rc) = 2 (2 Ct), where Ct is the Ct ß-actin – Ct ND1 [16].

### 2.4. Statistical Analysis

Continuous variables are expressed as the mean ± standard deviation. Logarithmic transformation was applied to the data showing non-normal distribution. Group comparisons were performed using the Student’s *t*-test and one-way ANOVA, followed by the least significant difference (LSD) test. *p* < 0.05 was considered statistically significant. General linear models were used to identify the independent predictors and for the adjustment of confounding factors for blood oxidative stress markers and mtDNA copy numbers between PD and non-PD groups. As the mtDNA copy number displayed a non-linear distribution pattern, we changed it to a delta Ct set for comparison, whereas oxidative stress markers displayed in a linear distribution. The contrast factor was applied in a one-way analysis of variances to test for linear trends displayed by the various ages, high–low dopamine dose, and quartile dopamine dose subgroups. Statistical analysis was performed using the Statistical Package for Social Science Program (SPSS for Windows, version 11.5; SPSS, Chicago, IL, USA).

## 3. Results

### 3.1. Differences between Demographics, Comorbidities, and Biological Markers of PD and Non-PD Control Cohorts

A higher average BMI and more subjects with a history of hypertension were noted in the non-PD control group than in the PD group. Otherwise, no significant differences between the two groups were found in terms of age, sex, or other medical conditions. However, a significantly higher average level of serum oxidant TBARS (1.59 ± 0.63 vs. 1.36 ± 0.76 μmol/L, *p* < 0.001), and a lower average level of serum antioxidant thiols (1.72 ± 0.42 vs. 1.79 ± 0.43 μmol/L, *p* < 0.001) were noted in the PD cohort as compared with the non-PD controls. The average mtDNA copy number in the blood cells was also found to be significantly lower in the PD cohort than in non-PD controls (2.40 ± 0.17 vs. 2.46 ± 0.22, *p* < 0.001). Differences between these biomarkers remained consistent after adjustment for age, sex, BMI, smoking, and other medical conditions (see Table 1).

### 3.2. Clarification of Confounding Factors Which Influence Expression of PD-Related Biomarkers

To clarify the reciprocal influences of these biomarkers and their relationships with age and sex, we conducted a bivariate correlative study. We identified aging-associated decreases of mtDNA copy number and antioxidative thiols levels, as well as an inverse correlation between thiols and TBARS levels. Other findings included a lower average mtDNA copy number in male subjects (Appendix A). To further clarify the confounding influence of age on the mtDNA copy number, TBARS levels, and thiols levels, we categorized all subjects into 5 subgroups according to their age at the time of the initial study. These age subgroups were as follows: Under 51 years; between 51 and 60; between 61 and 70; between 71 and 80; and above 81 years of age. We then compared the average mtDNA copy number, TBARS levels, and thiols levels of the corresponding age subgroups between the two cohorts. The results are shown in Figure 1. A relatively lower average mtDNA copy number was observed in all age subgroups of the PD cohort in comparison with the corresponding non-PD age subgroups, with particular significance in the age subgroups of patients under 51 years, 51 to 60, and 61 to 70, after post hoc analysis. In addition, while a notable trend of a decreased mtDNA copy number with increased age was observed in the non-PD cohort (*P_trend_* < 0.001), no similar result was observed in the PD cohort (Figure 1A). Moreover, while the non-PD group exhibited progressively increasing TBARS levels with age (*P_trent_* < 0.001), this trend was not observed in the PD group (Figure 1B). Age-related results were also noted in the average thiols levels (*P_trend_* < 0.001 for both groups): Although relatively lower thiols levels were found in the PD cohort, post hoc analysis revealed no significant differences between all age subgroups of the two cohorts (Figure 1C).

### 3.3. Changes of PD-Related Biomarkers during Follow-Up of PD Patients

All follow-up studies of these biomarkers between the two cohorts are shown in Figure 2. The percentages were arrived at by identifying the change in the mtDNA copy number, TBARS, or thiols between the first and second evaluations, which varied from 6 to 67 months. Any increase or decrease was then expressed as a percentage and plotted. We found that the mtDNA copy number in the PD cohort exhibited more significant reductions than in the non-PD controls during the study period (Figure 2A), which remained significant after adjustment for age, sex, and other factors (*p* < 0.001). The estimated average amount of copy number decrease for PD patient was 0.38% per month, whereas the decrease for non-PD controls was 0.25%. These data were acquired by dividing the average percentage of mtDNA copy number decrease by the average months of follow-up in PD and non-PD cohorts, respectively. Meanwhile, although the TBARS level in the non-PD controls was found to consistently increase during the study period, the PD group showed an initial reduction, which increased over time (Figure 2B). Although the serum thiols levels decreased with time in both cohorts, there was no significant difference between the two groups (Figure 2C).

### 3.4. Investigation for the Effects of Dopamine on Oxidative Stress and mtDNA Copy Number

To further investigate the potential factors implicated in the mtDNA copy number and TBARS changes between the age subgroups of PD and non-PD cohorts, we examined the related medical conditions and prescribed medications between the two groups. We identified dopamine and its agonists as the medication unique to the PD cohort. Therefore, at the time of blood withdrawal, we calculated the doses of prescribed dopamine and any dopamine agonists administered to patients to arrive at a TEDD for each PD patient. To establish the baseline study, patients were then categorized into two subgroups, higher or lower than 325 mg per day, according to their TEDD. This dose was determined by the average prescribed dose of all PD patients included in the study, whereas subjects in the non-PD control, never having been administered dopamine, were categorized as the dopamine-naive group. Our analysis revealed an association between lower mtDNA copy number and higher dopamine dose in PD patients, especially in the younger age subgroup (*P_trend_* = 0.046 and 0.271 for low and high groups, Figure 3A). The study for the TBARS level change also found an association between lower TBARS levels and higher dopamine dose in the younger age subgroup of PD patients (*P_trend_* = 0.002 and 0.047 for low- and high-dose groups, Figure 3B). No similar finding was observed in the change of thiols levels in either of the dopamine groups, or the control group (Figure 3C).

### 3.5. Effects of Continuous Dopamine Therapy on PD-Related Biomarkers

The effects of continuous dopamine therapy on changes of the PD-related biomarkers were investigated and shown in Figure 4. The four groups of patients based on the quartile categorization of the prescribed dopamine dose are as follows: <350; 350–550; 550–750; and >750 mg. The dopamine dose chosen for the data set was the dose prescribed at the last follow-up. Analysis revealed that TBARS levels showed initial decreases in all four quartiles, which grew milder over time, albeit at different stages of the follow-up period. A clear association between the dopamine dose and TBARS level change was noted, in which the higher dopamine dose quartile exhibited only a mild decrease in the TBARS level. No significant correlative effects of dopamine dose on changes to mtDNA copy number or thiols levels were noted.

## 4. Discussion

In recent decades, the unraveling of the mechanisms regulating to mitochondrial biogenesis have enhanced our understanding of how, through increasing replication and transcription of mitochondrial DNA, cells exert themselves to overcome environmental challenges and maintain cellular life [17]. Failure of this process results in progressive cellular dysfunction and death, which in turn could cause organ abnormalities and generation of disease [18]. In the present study, we identified a significantly lower mtDNA copy number in the peripheral blood cells of PD patients in comparison with the non-PD controls, which is consistent with previous reports [10,11,12]. The lower mtDNA copy numbers, found in PD patients, were noted to also present with higher oxidative TBARS and lower antioxidative thiols in comparison with all corresponding age subgroups of non-PD controls. Commonly, biogenesis reacts to disease pathology through increasing the mtDNA copy number [18]; however, this was not observed in the baseline study of our PD patients. Moreover, during the follow-up study period, we noted a more rapid decrease in the mtDNA copy number in the PD group as compared to the non-PD controls. Although a decrease in mtDNA copy number has been noted to be associated with the aging process, the more rapid decrease in PD patients indicated a systemic failure of cellular biogenesis. Whether these findings resulted directly from malfunction/dysfunction of biogenesis, or the failure of other mito-nuclear communication pathways, including anterograde regulation or the retrograde response, remains unclear and awaits further study [19,20]. Our study provides the first results from longitudinal human observation which are consistent with previous findings from cellular and animal model studies [21,22]. The lower mtDNA copy number and more rapid decrease over time in PD patients than non-PD controls (0.37% vs. 0.24% per month, respectively) offers the potential to be used as a biomarker in future clinical applications in both pre- and post-PD diagnosis periods.

Human serum TBARS levels are known to increase naturally as part of the aging process [23]. The findings of our baseline study of non-PD controls were consistent with this. However, elevated TBARS levels may also be associated with cellular stress, often triggered by disease pathogenesis. In our baseline study, we found consistently higher levels of TBARS in all age subgroups of PD patients in comparison with the non-PD controls. Interestingly, the level of TBARS exhibited no increase among the older age subgroups of PD patients. Additionally, we identified an association between lower TBARS levels and higher doses of dopamine therapy, indicating that the TBARS level in PD patients may be influenced by not only age but also dopamine administration. This correlation between dopamine administration and TBARS level was also found to be accompanied by a milder decrease in mtDNA copy number in the older age subgroup of PD patients in comparison with the non-PD controls. It is plausible that these findings result from alterations to mitochondrial biogenesis, due to the attenuation of oxidative stress by dopamine therapy [24].

The finding of a decrease in levels of human serum TBARS as a result of dopamine therapy, likely due to the attenuation of oxidative stress, has not previously been reported in large-scale human studies. Dopamine has traditionally been viewed as a double-edged sword, providing therapeutic effects while also causing oxidative stress [25,26]. Comprehensive studies have found low doses of dopamine offer neuro-protective effects, but may lead to cellular death through oxidative stress-induced apoptosis under higher doses [27,28,29,30]. Our present study shows a consistent decrease in TBARS levels due to dopamine administration from a baseline cross-sectional study and longitudinal follow-up study. These findings indicate that dopamine may provide protective effects to PD patients through the attenuation of oxidative stress, at least under lower doses.

The findings of the follow-up study were consistent with those of the baseline study. The TBARS levels in PD patients exhibited an initial decrease, which became milder with time (Figure 2B). Moreover, investigation into the effects of different doses of dopamine on TBARS levels found that the lower dose quartiles exhibited more significant decreases compared to the higher dose quartiles (Figure 3B). Furthermore, no trends were noted in either the mtDNA copy number or thiols levels throughout the follow-up study period.

The decision to select TBARS and thiols as the biomarkers used to measure oxidative stress in this study was based on our previous observations of their reciprocal relationships with the mtDNA copy number, in addition to the reliability and consistent reproducibility of these biomarkers in clinical practice. It is important to note, however, that due to the possible influence of the ferroptosis process on TBARS levels, it may be helpful to check the serum iron level for the potential confounding influence of ferroptosis on elevated TBARS levels [31]. In addition, although measurements of TBARS and thiols can provide informative data, future investigations may consider measurement of additional parameters related to reactive oxygen species and antioxidant pathways to provide further support and strengthen study results. Furthermore, this study did not distinguish between dopamine and dopamine agonist. Thus, additional research may be required to clarify any potential differences on oxidative stress [32].

Taken together, the findings of the baseline and follow-up studies herein identified sequential alterations to blood cell mtDNA copy number and serum markers of oxidative stress in PD patients. These biomarkers were further found to be influenced by dopamine administration. Our study provides support for the use of these biomarkers in clinical application to monitor both the progress of PD and the influence of dopamine therapy. Authors should discuss the results and how they can be interpreted in the perspective of previous studies and of the working hypotheses. The findings and their implications should be discussed in the broadest context possible. Future research directions may also be highlighted.

## Figures and Tables

**Figure 1 antioxidants-09-01159-f001:**
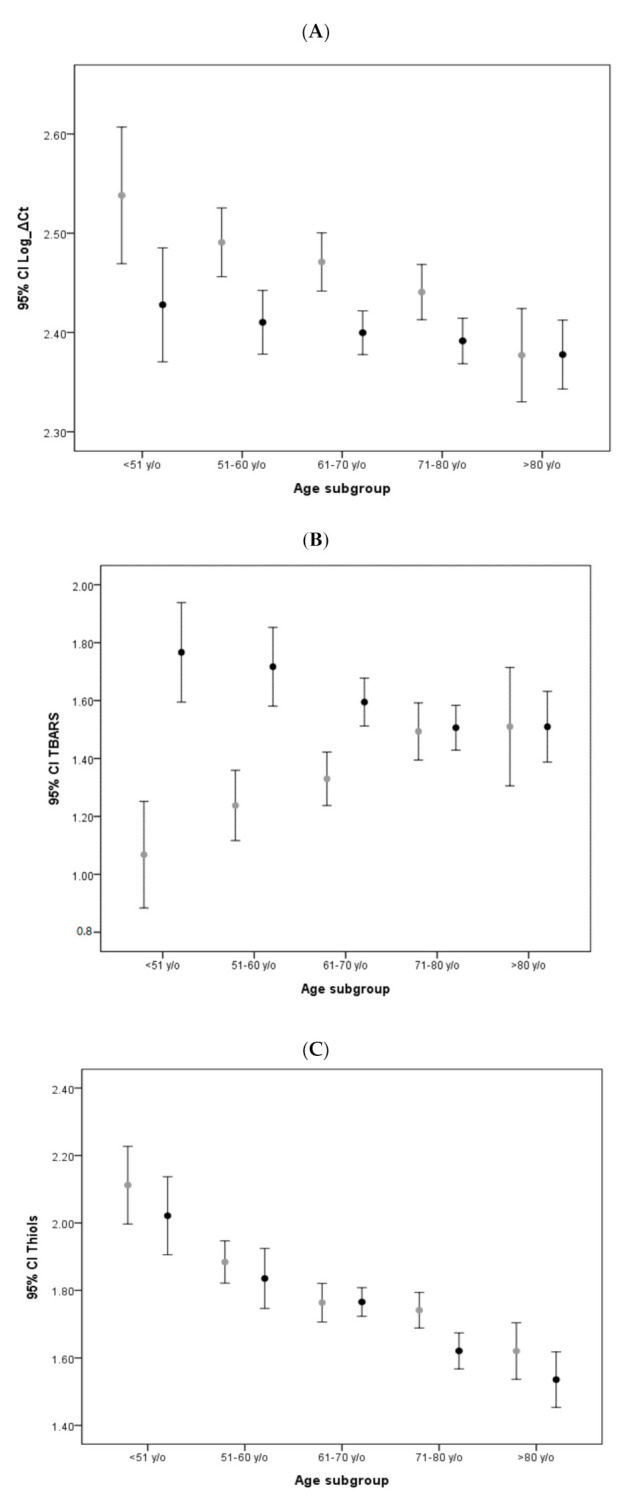
Age-stratified comparison of oxidative stress markers and mtDNA copy number between PD patients and non-PD controls: Error bar plots show levels of average LogΔCt copy number (**A**), TBARS (**B**), and thiols (**C**) in different age subgroups of PD patients and non-PD controls. (● in (**A**–**C**) represents PD patient group; ● in (**A**–**C**) represents non-PD control group).

**Figure 2 antioxidants-09-01159-f002:**
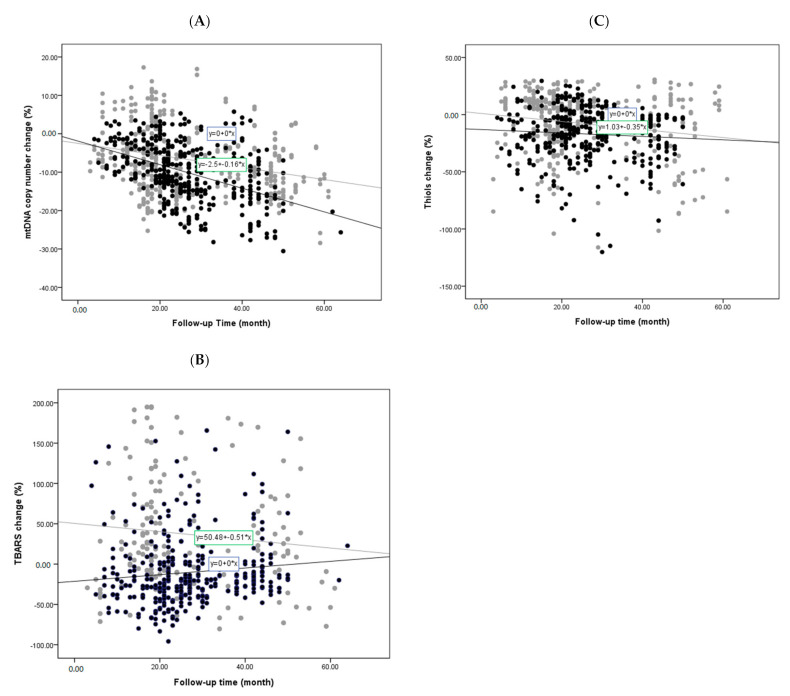
Changes of oxidative stress markers and mtDNA copy number during follow-up studies of PD patients and non-PD controls: Simple scatter plots show changes of LogΔCt copy number (**A**), TBARS (**B**), and thiols (**C**) during follow-up studies of PD patients and non-PD controls. (● in (**A**–**C**) represents PD patient group; ● in (**A**–**C**) represents non-PD control group).

**Figure 3 antioxidants-09-01159-f003:**
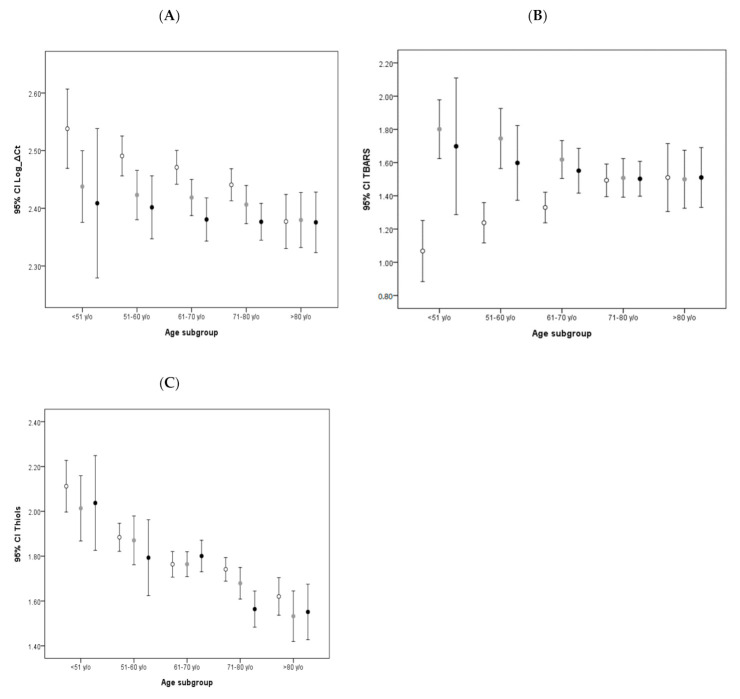
Age-stratified comparison of influences of various total equivalent dopamine doses (TEDDs) on oxidative stress markers and mtDNA copy number: Error bar plots show levels of average Log Ct copy number (**A**), TBARS (**B**), and thiols (**C**) in different TEDD subgroups of PD patients and non-PD controls. (● in (**A**–**C**) represents higher dopamine dose group of PD patients; ● in (**A**–**C**) represents lower dopamine dose group of PD patients; ○ in (**A**–**C**) represents dopamine-naive group of non-PD controls).

**Figure 4 antioxidants-09-01159-f004:**
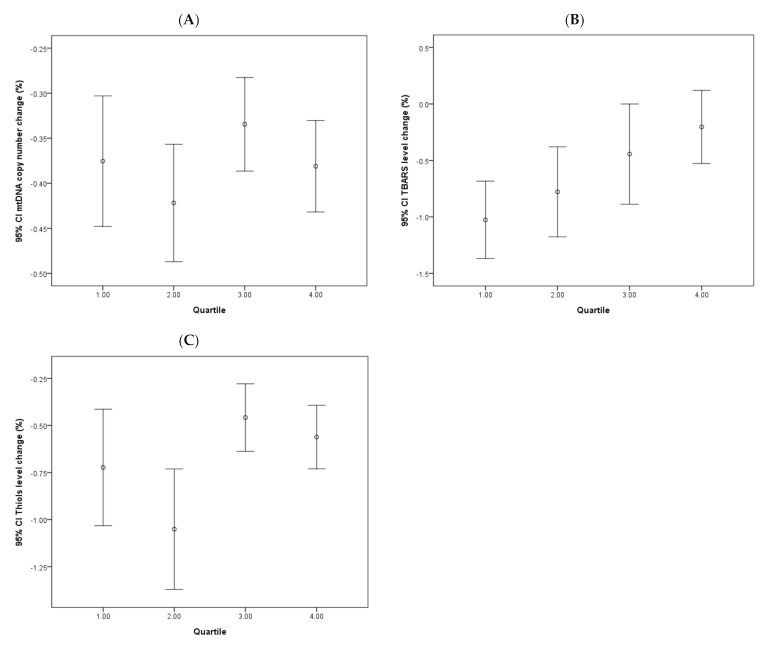
Changes of continuous dopamine administration on various oxidative stress markers and mtDNA copy number in PD patients. Error bar plots show levels of average LogΔCt copy number (**A**), TBARS (**B**), and thiols (**C**) in different quartile subgroups of dopamine dose in PD patients.

**Table 1 antioxidants-09-01159-t001:** Demographics, mitochondrial DNA (mtDNA) copy number, thiobarbituric acid reactive substances (TBARS), and thiols of Parkinson’s disease (PD) patients and non-PD controls.

Variable	PD Patients	Non-PD Controls	*p*
	(*n* = 725)	(*n* = 744)	
Age, year (SD)	67.1 (11.1)	66.2 (9.4)	0.109
Male, number (%)	396 (54.6)	403 (54.2)	0.861
BMI, mean (SD)	24.33 (3.64)	24.97 (3.46)	<0.001
Medical history			
Hypertension (%)	335 (46.2)	411 (55.2)	<0.001
Diabetes mellitus (%)	139 (19.2)	147 (19.8)	0.787
Smoking (%)	120 (16.3)	134 (18.0)	0.511
mtDNA copy number, log Delta Ct (SD)	2.40 (0.17)	2.46 (0.22)	<0.001
TBARS, μmol/L (SD)	1.59 (0.63)	1.36 (0.76)	<0.001
Thiols, μmol/L (SD)	1.72 (0.42)	1.79 (0.43)	<0.001

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
