# Peer review of "Dopamine Therapy and the Regulation of Oxidative Stress and Mitochondrial DNA Copy Number in Patients with Parkinson’s Disease"

_antioxidants, 2020, doi:10.3390/antiox9111159_

Round 1
Reviewer 1 Report
Manuscript details: Manuscript ID: antioxidants-968271
Antioxidants
Type of manuscript: Article
Title: Dopamine Therapy and The Regulation of Oxidative Stress and
Mitochondrial DNA Copy Number in Patients with Parkinson’s Disease
Revision:
The Article entitled: “Dopamine Therapy and The Regulation of Oxidative Stress and Mitochondrial DNA Copy Number in Patients with Parkinson’s Disease” is very interesting.
Authors described as biological markers and in particular mitochondrial DNA copy number in the peripheral blood cells, serum TBARS and thiols levels, change in long term in patients with Parkinson disease (PD) versus patients non- Parkinson disease; they correlated these parameters with age and sex and studied the effect of dopamine therapy on these biomarkers in PD patients.
I think that the manuscript is suitable for its publication in “Antioxidants” after minor revisions:
Minor revisions
Review the text and correct it, because:
- in the abstract section mitochondrial DNA copy number is abbreviated with mtCN, but in all the rest of the text this abbreviation does not appear, in fact mitochondrial DNA copy number appears as mtDNA copy number (line 44, 105, 127,158....etc)
- Materials and methods:
- in the section 2.3. line 88: convert the title "Assessment of oxidative and antioxidative stress capacities" in Assessment of serum total thiols and thiobarbituric acid reactive substances" and write below, when you mention them, that one is an antioxidant parameter, while the other is a parameter that is produced following oxidative stress. You can also write "Assessment of antioxidant and oxidant markers". In your title you speak about antioxidant stress that does not exist.
- Results
- The caption of the table is missing
- The p value for mtDNA copy number and TBARS is very low for very similar mean and standard deviation values: perhaps because the number of samples is very high?
- In figure 2A: there is an incongruence with the caption and the Y value. Verify it.
Author Response
Point-to-Point Response to the Reviewers’ Critiques
REVIEWER 1:
Comments and Suggestions for Authors
The Article entitled: “Dopamine Therapy and The Regulation of Oxidative Stress and Mitochondrial DNA Copy Number in Patients with Parkinson’s Disease” is very interesting.
Authors described as biological markers and in particular mitochondrial DNA copy number in the peripheral blood cells, serum TBARS and thiols levels, change in long term in patients with Parkinson disease (PD) versus patients non- Parkinson disease; they correlated these parameters with age and sex and studied the effect of dopamine therapy on these biomarkers in PD patients.
I think that the manuscript is suitable for its publication in “Antioxidants” after minor revisions:
Minor revisions
Review the text and correct it, because:
- in the abstract section mitochondrial DNA copy number is abbreviated with mtCN, but in all the rest of the text this abbreviation does not appear, in fact mitochondrial DNA copy number appears as mtDNA copy number (line 44, 105, 127,158....etc)
Response:
Thank you for the notification. Our initial purpose for using this abbreviated term was to comply with the submission guidelines that restricted words to a maximum of 200 in the Abstract. In the present revised manuscript, we have cancelled the abbreviated term “mtCN” and returned to the original full term of “mitochondrial DNA copy number” in the Abstract. Additionally, we have made a minor correction of the Abstract by deleting the word: “the” and maintained a word count of less than 200.
2 Materials and methods in the section 2.3. line 88: convert the title "Assessment of oxidative and antioxidative stress capacities" in Assessment of serum total thiols and thiobarbituric acid reactive substances" and write below, when you mention them, that one is an antioxidant parameter, while the other is a parameter that is produced following oxidative stress. You can also write "Assessment of antioxidant and oxidant markers". In your title you speak about antioxidant stress that does not exist.
Response:
Thank you for your insightful advice. We have changed the title to, “Assessment of antioxidant and oxidant markers”. Furthermore, we have adjusted the text to be more clear about thiols being an antioxidant parameter and TBARS being produced following oxidative stress.
3 Results The caption of the table is missing
Response:
Thank you for the notification. We experienced some minor formatting issues, and have amended that section accordingly.
- The p value for mtDNA copy number and TBARS is very low for very similar mean and standard deviation values: perhaps because the number of samples is very high?
Response:
Thank you for the query. You are correct. It is actually a high p value (<0.001) due to the high sample number and narrow standard deviation. Besides, we compared the mtDNA copy number as log due to its non-linear distribution. This may also make for a high significant p value.
Classification Tablea,b |
|||||
|
Observed |
Predicted |
|||
|
PD |
Percentage Correct |
|||
|
.00 |
1.00 |
|||
Step 0 |
PD |
.00 |
744 |
0 |
100.0 |
1.00 |
725 |
0 |
.0 |
||
Overall Percentage |
|
|
50.6 |
||
a. Constant is included in the model. |
|||||
b. The cut value is .500 |
Variables in the Equation |
|||||||
|
B |
S.E. |
Wald |
df |
Sig. |
Exp(B) |
|
Step 1a |
Age |
.005 |
.005 |
.786 |
1 |
.375 |
1.005 |
Sex |
-.011 |
.108 |
.010 |
1 |
.921 |
.989 |
|
Log_ΔCt |
-1.605 |
.274 |
34.302 |
1 |
.000 |
.201 |
|
Constant |
3.537 |
.799 |
19.618 |
1 |
.000 |
34.366 |
|
Step 2a |
Age |
.005 |
.005 |
.808 |
1 |
.369 |
1.005 |
Log_ΔCt |
-1.604 |
.273 |
34.391 |
1 |
.000 |
.201 |
|
Constant |
3.524 |
.787 |
20.051 |
1 |
.000 |
33.903 |
|
Step 3a |
Log_ΔCt |
-1.633 |
.272 |
36.162 |
1 |
.000 |
.195 |
Constant |
3.909 |
.661 |
34.988 |
1 |
.000 |
49.858 |
|
a. Variable(s) entered on step 1: Age, Sex, Log_ΔCt. |
Classification Tablea,b |
|||||
|
Observed |
Predicted |
|||
|
PD |
Percentage Correct |
|||
|
.00 |
1.00 |
|||
Step 0 |
PD |
.00 |
744 |
0 |
100.0 |
1.00 |
725 |
0 |
.0 |
||
Overall Percentage |
|
|
50.6 |
||
a. Constant is included in the model. |
|||||
b. The cut value is .500 |
Variables in the Equation |
|||||||
|
B |
S.E. |
Wald |
df |
Sig. |
Exp(B) |
|
Step 1a |
Age |
.007 |
.005 |
1.854 |
1 |
.173 |
1.007 |
Sex |
.024 |
.108 |
.049 |
1 |
.825 |
1.024 |
|
TBARS |
.468 |
.078 |
35.865 |
1 |
.000 |
1.597 |
|
Constant |
-1.228 |
.378 |
10.560 |
1 |
.001 |
.293 |
|
Step 2a |
Age |
.007 |
.005 |
1.815 |
1 |
.178 |
1.007 |
TBARS |
.469 |
.078 |
36.120 |
1 |
.000 |
1.599 |
|
Constant |
-1.209 |
.369 |
10.769 |
1 |
.001 |
.298 |
|
Step 3a |
TBARS |
.473 |
.078 |
36.781 |
1 |
.000 |
1.605 |
Constant |
-.744 |
.126 |
34.700 |
1 |
.000 |
.475 |
|
a. Variable(s) entered on step 1: Age, Sex, TBARS. |
- In figure 2A: there is an incongruence with the caption and the Y value. Verify it.
Response:
Thank you for the observation. We have changed the caption of Figure 2A from “Log Ct copy number” to “mtDNA copy number”. We have verified that the figures and caption are consistent.
Reviewer 2 Report
The authors studied dopamine therapy in PD patients, the involvement of oxidative stress and mitochondrial DNA copy number. Decreased mtCN and anti-oxidant thiols level, but elevated oxidative TBARS level presented in PD patients. Further dopamine administration lower serum TBARS. There appears to be an indirect relationship with mtCN, where a decrease in TBARS was found to diminish the effect 25 of pathogenetic and age-related decrease in mtCN in PD patients. Overall, the manuscript is well written, and the data support the conclusions made by the authors. However, a few points should be clarified prior to publication.
- It would be good to check the serum iron level, since increased lipid peroxidation associated with ferroptosis.
- The authors should measure additional parameters related to ROS and antioxidant pathways, which is strengthen the story.
Author Response
Point-to-Point Response to the Reviewers’ Critiques
REVIEWER 2:
Comments and Suggestions for Authors
The authors studied dopamine therapy in PD patients, the involvement of oxidative stress and mitochondrial DNA copy number. Decreased mtCN and anti-oxidant thiols level, but elevated oxidative TBARS level presented in PD patients. Further dopamine administration lower serum TBARS. There appears to be an indirect relationship with mtCN, where a decrease in TBARS was found to diminish the effect 25 of pathogenetic and age-related decrease in mtCN in PD patients. Overall, the manuscript is well written, and the data support the conclusions made by the authors. However, a few points should be clarified prior to publication.
- It would be good to check the serum iron level, since increased lipid peroxidation associated with ferroptosis.
Response:
I truly appreciate your insightful suggestion. We agree with the importance of ferroptosis in generation of human neurodegenerative diseases and its mutual and causal relationships with the serum iron and lipid peroxidation levels. For this, we will add one sentence and a reference to the Discussion section to explain further. Due to various issues, including IRB and time constraints, we would be unable to carry out the necessary experiments. However, the purpose of our present study was to seek simple biomarkers as tools which can be reliable and reproducible for monitoring the progression of Parkinson disease and patient response to dopamine therapy. We will also remind the reader or other researchers of the importance of the serum iron level in possibly influencing the lipid peroxidation in the monitoring of Parkinson disease to avoid any possible bias.
From line 272 to 274: It is important to note, however, that due to the possible influence of the ferroptosis process on TBARS levels, it may be helpful to check the serum iron level for the potential confounding influence of ferroptosis on elevated TBARS levels [31].
- The authors should measure additional parameters related to ROS and antioxidant pathways, which is strengthen the story.
Response:
Thank you for the suggestion. We hope the response to the previous issue will also be applicable here. While there are additional parameters which could be used to monitor the burden of oxidative stress in humans, we have previously studied thiols and TBARS and have confirmed the relationship with mtDNA copy number. We have confidence in this association, so we selected these as biomarkers. We mention this in the second to last paragraph of the Discussion section. However, for a clearer description, we will add one more sentence to notify reader of the potential of strengthening our findings if additional parameters related to ROS and antioxidant pathways were measured.
From line 274 to 277: In addition, although measurements of TBARS and thiols can provide informative data, future investigations may consider measurement of additional parameters related to ROS and antioxidant pathways to provide further support and strengthen study results.

Reviewer 3 Report
Summary
Chen et al present data on metabolism in Parkinson's Disease patients and controls. The authors use mitochondrial DNA copy number to measure mitrochondria abundance, thiobarbituric acid compounds and thiols to measure oxidative stress. The authors report a negative correlations between Parkinson's Disease and mitochondria DNA copy number and thiols. The authors also report a positive correlation between Parkinson's Disease and thiobarbituric acid compounds.
Main comments:
Introduction: The introduction is a little difficult to follow. This is primarily due to word choice that creates 'indirect' logic flow. For example, the first paragraph describes that mtDNA increases to increase mitochondrial respiration. However, the next sentence (line 45) does not mention mitochondria, mtDNA, or respiration. Instead the sentence refers to "...failure of this process...". I recommend changing the sentence to "...failure to increase mitochondrial respiration...".
Another example: the second paragraph ends by stating "...finding direct evidence of this by retrieving tissue samples from living brains is problematic. We herein report the results of our investigation into the long-term changes of oxidative stress and mtDNA copy number, and the probable effects of continuous dopamine therapy on these biomarkers in PD patients." This wording makes it seem like the authors will report on samples collected from the brain. This is misleading. The authors should make it clear that the data is from the periphery.
It would also be helpful if the second paragraph stated a hypothesis or expected results.
Analysis: All of the data collection and analysis appears appropriate. However, I am concerned about the statistics. The analysis contains multiple comparisons that were decided before data were collected. These multiple comparisons are not corrected with standard post hoc tests. There are several tests that would be appropriate (Tukeys, SNK, Bonferroni).
Data: Most of the data are well presented and convincing. I am concerned about the Thiol data. The effect is very weak and significance likely would not survive a post hoc analysis (see above). Therefore, further analysis is necessary. At a minimum the effect size should be calcluated (i.e. thiol concentration in PD vs Control; Cohens D or Glasses D).
Minor Comments:
Table 1 should be one page.
Figure 2 is small and hard to read. Suggest making each graph larger and increase line and text size (i.e. the point of lines and the font size of the text).
There are several small text errors. Below are two examples.
For example (line 73): "Venous blood samples were collected after an overnight fasting. " Should be either "..after an overnight fast." or "...after fasting overnight."
(line 86): change to "A TBARS standard curve was.."
Author Response
Point-to-Point Response to the Reviewers’ Critiques
REVIEWER 3:
Comments and Suggestions for Authors
Summary
Chen et al present data on metabolism in Parkinson's Disease patients and controls. The authors use mitochondrial DNA copy number to measure mitochondrial abundance, thiobarbituric acid compounds and thiols to measure oxidative stress. The authors report a negative correlations between Parkinson's Disease and mitochondria DNA copy number and thiols. The authors also report a positive correlation between Parkinson's Disease and thiobarbituric acid compounds.
Main comments:
- Introduction: The introduction is a little difficult to follow. This is primarily due to word choice that creates 'indirect' logic flow. For example, the first paragraph describes that mtDNA increases to increase mitochondrial respiration. However, the next sentence (line 45) does not mention mitochondria, mtDNA, or respiration. Instead the sentence refers to "...failure of this process...". I recommend changing the sentence to "...failure to increase mitochondrial respiration...".
Response:
Thank you for the suggestion. We apologize for any confusion and will try to make the flow of the text more logical.
- Another example: the second paragraph ends by stating "...finding direct evidence of this by retrieving tissue samples from living brains is problematic. We herein report the results of our investigation into the long-term changes of oxidative stress and mtDNA copy number, and the probable effects of continuous dopamine therapy on these biomarkers in PD patients." This wording makes it seem like the authors will report on samples collected from the brain. This is misleading. The authors should make it clear that the data is from the periphery.
Response:
Thank you for the suggestion. We have altered the paragraph to clarify our methods and hypothesis.
Line54-56: However, as finding direct evidence of this by retrieving tissue samples from living brains is problematic, we thus hypothesized that this serial cascade response may be detected by monitoring changes in human blood samples. We herein report the results of our investigation into the baseline and long-term changes of oxidative stress and mtDNA copy number, and the probable effects of continuous dopamine therapy on these biomarkers in PD patients.
- It would also be helpful if the second paragraph stated a hypothesis or expected results.
Response:
Thank you, the amendments to the Introduction section may be referenced above.
- Analysis: All of the data collection and analysis appears appropriate. However, I am concerned about the statistics. The analysis contains multiple comparisons that were decided before data were collected. These multiple comparisons are not corrected with standard post hoc tests. There are several tests that would be appropriate (Tukeys, SNK, Bonferroni).
Response:
Thank you for the notification. We are sorry for not mentioning the statistical method used for pos-hoc analysis. We have added this to the statistical analysis paragraph in the Methods section. Additionally, we have provided the data or description of post hoc analysis in some necessary parts of the Results.
Line 115-116: Group comparisons were performed using the Student’s t-test and one-way ANOVA followed by the least significant difference (LSD) test.
Line 148-151: A relatively lower average mtDNA copy number was observed in all age subgroups of the PD cohort in comparison with the corresponding Non-PD age subgroups, with particular significance in the age subgroups of patients 30 to 40, and 40 to 50, after post-hoc analysis.
- Data: Most of the data are well presented and convincing. I am concerned about the Thiol data. The effect is very weak and significance likely would not survive a post hoc analysis (see above). Therefore, further analysis is necessary. At a minimum the effect size should be calculated (i.e. thiol concentration in PD vs Control; Cohens D or Glasses D).
Response:
Thank you for the observation. The thiols data was actually insignificant after post hoc analysis. We have corrected it in the new manuscript.
Line 154-157: Age-related results were also noted in the average thiols levels (Ptrend < 0.001 for both groups), although relatively lower thiols levels were found in the PD cohort, post hoc analysis revealed no significant differences between all age subgroups of the two cohorts (Fig 1C).
Minor Comments:
- Table 1 should be one page.
Response:
Thank you, we agree that the Table should be consistent, and we will ask the Editor to address this before the paper is published.
- Figure 2 is small and hard to read. Suggest making each graph larger and increase line and text size (i.e. the point of lines and the font size of the text).
Response:
Thank you for the suggestion. We have made a new graph accordingly. We hope this will more suitably express the data.
- There are several small text errors. Below are two examples.
For example (line 73): "Venous blood samples were collected after an overnight fasting. " Should be either "..after an overnight fast." or "...after fasting overnight."
(line 86): change to "A TBARS standard curve was.."
Response
We apologize for these errors and have corrected them accordingly. We have also proofread other parts to ensure accurate text content.
